# Dietary Resveratrol Alleviates AFB1-Induced Ileum Damage in Ducks via the Nrf2 and NF-κB/NLRP3 Signaling Pathways and CYP1A1/2 Expressions

Hao Yang, Yingjie Wang, Chunting Yu, Yihan Jiao, Ruoshi Zhang, Sanjun Jin and Xingjun Feng *

Laboratory of Molecular Nutrition, Institute of Animal Nutrition, Northeast Agricultural University, Changjiang Street 600#, Xiangfang District, Harbin 150030, China; S190501046@neau.edu.cn (H.Y.); wangyingjie@neau.edu.cn (Y.W.); S210501037@neau.edu.cn (C.Y.); S210501804@neau.edu.cn (Y.J.); A05180171@neau.edu.cn (R.Z.); S210501043@neau.edu.cn (S.J.)
* Correspondence: fengxingjun@neau.edu.cn; Tel.: +86-451-55191395

**Abstract:** The aim of this study was to explore the mechanism underlying the protective effects of resveratrol against Aflatoxin B1-induced ileum injury in ducks. A corn–soybean meal-basal diet and two test diets (500 mg/kg resveratrol +0.2 mg Aflatoxin B1/kg, 0.2 mg $AFB_1$/kg) were used in a 10-wk design trial ($n = 15$ ducks/group). These results showed that the toxicity of Aflatoxin B1 significantly reduced the antioxidant capacity of duck ileum and induced inflammation, oxidative stress, mitochondrial dysfunction and DNA damage in ducks. The expression of genes, including CYP1A2, CYP2A6, and CYP3A4, at the mRNA level was significantly upregulated ($p < 0.05$) by $AFB_1$. The level of Nrf2 was suppressed ($p < 0.05$) and the mRNA and protein level of NF-κB was activated ($p < 0.05$) in the $AFB_1$ group. However, supplementation with 500 mg/kg dietary resveratrol in Aflatoxin B1-induced ducks significantly ameliorated these alterations and decreased the mRNA expression of CYP1A1 and CYP1A2 ($p < 0.05$) and the production of $AFB_1$-DNA adducts ($p < 0.05$). The results proved that resveratrol alleviated ileum injury induced by AFB1, decreased the production of $AFB_1$-DNA adducts by downregulating the expression of CYP1A1 and CYP1A2, and reduced DNA damage and oxidative stress via the Nrf2/ Keap1 and NF-κB/NLRP3 signaling pathways.

**Keywords:** mitochondrial dysfunction; anti-inflammatory; DNA damage; antioxidant; cytochrome P450





## 1. Introduction

Aflatoxins (AFs) are food-borne secondary toxic metabolites produced by *Aspergillus flavus* [1]. Aflatoxin pollution of animal feed has imposed great threats to the breeding industry and feed industry. Maize is widely used as one of the most major energy feeds for poultry and easily contaminated by Aspergillus flavus. Aflatoxins have serious toxic effects on humans and animals. Among the various AFs, Aflatoxin $B_1$ ($AFB_1$) is the most toxic and critical. The toxicity of $AFB_1$ is much higher than that of cyanide, arsenide and organic pesticides. It was listed as a class I strong carcinogen in 2010 and poses a serious carcinogenicity in humans and animals. Multiple reports are available regarding $AFB_1$, which causes various negative effects on animal growth and development such as reductions in growth rate, malnutrition, immune dysregulation, and gastrointestinal disturbances [2]. Lipid peroxidation, DNA damage and excessive reactive oxygen species (ROS) are the major manifestations of aflatoxicosis [3]. Past studies reported that $AFB_1$ was primarily activated by the cytochrome P450 (CYP450) in the liver and subsequently bound with nucleophilic sites in cell's DNA and formed $AFB_1$-DNA adducts [4]. Modern medicine believes that $AFB_1$-DNA adducts in plasma are a class of reliable biomarkers of DNA damage.

It is imperative to develop an effective antidote in order to reduce the hepatotoxicity of $AFB_1$ to humans and animals. Many studies have reported that plant polyphenols

possess antioxidant activity and detoxification functions, and can be used as physiological antioxidants and antidotes [5]. Some active extracts in plants have protective effects, including enhancing immunity, antioxidant activity and regulation of intestinal health [6]. Resveratrol (RES) (3,4,5 trihydoxystilbene), a natural plant extract, is distributed in various plant species, such as grape, peanut and Chinese knotweed. Since the "Mediterranean diet" was reported as a healthy dietary style in humans because the dietary pattern involves a series of antioxidants (especially RES from red wine) with a lower risk of cancer and cardiovascular diseases in 2001, the biological properties and health benefits of RES have received intensive attention [7]. Both in vivo and in vitro experiments also suggested that RES has many biological properties such as anti-inflammatory, antiapoptotic, anticancer, lipid regulatory and immunomodulatory properties, to alleviate numerous diseases and stress both in humans and animals [8–10]. RES has been demonstrated to play a vital role in preventing digestive system damage including irradiation-induced ileum damage and ileum ischemia-reperfusion injury in rats and ducks [11,12].

Previous studies have shown that the mRNA expression of nuclear Factor E2-related Factor 2 (Nrf2) and a series of downstream genes was decreased by $AFB_1$ in bursa of Fabricius of broiler chickens [13]. Nrf2 is one of the important signaling pathways that exerts the antioxidative effects. In addition, the Nrf2/ARE signaling pathway was suggested, which could be activated by RES dimer and RES monomer and which plays an important role in decreasing oxidative stress [14]. RES can downregulate the production of malondialdehyde (MDA) and induce oxidative stress via modulating the Nrf2 signaling pathway [15]. Dietary RES supplementation improves the nuclear translocation of Nrf2 and increases the levels of several phase-II detoxifying enzymes and antioxidant enzymatic protection system in primary rat hepatocytes [16]. $AFB_1$ can increase the expression of nuclear factor kappa B (NF-κB) and lead to the inflammatory reaction in vivo and in vitro [17]. NF-κB is a transcription factor activated by Toll-like receptor 4 (TLR4) in animals. In the subsequent inflammatory cascade, NF-κB is a key regulator which is responsible for mediating downstream inflammatory genes. Inflammasome Nod-like receptor family pyrin domain containing 3 (NLRP3) is regarded as a key target gene in inflammatory response. NLRP3 can be activated by the NF-κB signaling pathway and lead to the production of inflammatory cytokines in the immune system [18]. RES possesses anti-inflammatory activity probably by inhibiting the NF-κB expression caused by TLR4- regulated signaling in vitro [19].

Many previous studies have reported that $AFB_1$ is a potent hepatotoxic and hepatocarcinogenic mycotoxin. Nevertheless, there is evidence of $AFB_1$ potential toxicity in the digestive system. Before entering the liver, most $AFB_1$ is absorbed by the proximal small intestine, damages the intestinal morphology and causes dysfunction of the intestinal tract [20]. It has been proven that aflatoxins can induce ileum pathological damage in poultry, such as reductions in the length and width of ileum villus [21]. Among all poultry species, ducks are the most susceptible to $AFB_1$ because ducks have a limited ability to metabolize aflatoxins in feed [2]. To our knowledge, there have been no reports about the therapeutic effects of dietary RES on $AFB_1$-induced ileal damage in ducks. The effects of dietary RES on $AFB_1$-induced duck ileum damage were explored to support the health of ducks and reduce the economic loss of animal husbandry caused by $AFB_1$. The possible molecular mechanism of dietary RES was investigated based on the regulation of the Nrf2 and NF-κB/NLRP3 signaling pathways and CYP450 enzyme expression.

## 2. Materials and Methods

### 2.1. Materials

RES (purity $\geq$ 98%, CAS: 501–36–0) was bought from Nanjing Nutri-herb Biotech Co., Ltd. (Nangjing, China). $AFB_1$ (purity $\geq$ 98%, CASNO.1162–65–8) was bought from Shanghai Yuanye Bio-Technology Co., Ltd. (Shanghai, China).



### 2.2. Ethical Issues

All the procedures of laboratory animal were performed in with the Ethical and Animal Welfare Committee of China's Heilongjiang Province.

### 2.3. Ducks and Husbandry

All the ducks were allowed ad libitum access to water and feed with enough space. Forty-five 1-day-old male specific pathogen-free (SPF) ducks (Anas platyrhynchos, initial mean body weight $33.8 \pm 0.2$ g) were randomly assigned to 3 groups: CON, $AFB_1$, and RES+ $AFB_1$. There are 15 ducks in each group. The trial period lasted for 70 days. Ingredients and nutrient contents of basal diet in this study were formulated based on the National Research Council (1994) (shown in Supplemental Table S1). The RES+ $AFB_1$ group was fed a RES enriched diet in doses of 500 mg/kg and supplemented with 0.2 mg/kg $AFB_1$. The $AFB_1$ group was fed a basal diet with 0.2 mg/kg $AFB_1$ added. RES and $AFB_1$ were mixed in the feed by powder form.

### 2.4. Sample Collection

After fasting for 12 h, eight ducks in three groups were randomly selected for sample collection. Samples of duck blood were obtained from the jugular vein in anticoagulation tubes containing heparin (20 IU/mL) and centrifuged under cryogenic conditions for 5 min. The serum samples were obtained and were preserved in sterile tubes at $-80\,^{\circ}\mathrm{C}$ for further evaluation of enzymes and other biochemical markers in blood. Ducks were anesthetized by exposure to ether for 3 to 4 min and slaughtered after blood collection. The ileal sample for tissue histopathology analysis was kept in 4% paraformaldehyde solution. The tissue samples for ultrastructural morphology analysis were soaked to 2.5% glutaraldehyde. Remaining ileal tissue was stored frozen and prepared for subsequent analysis stages.

### 2.5. Assay of Blood $AFB_1$-DNA Adduct Levels

The generation of $AFB_1$-DNA adducts in serum was determined with 10% fresh serum from ducks. All measurement steps were completed according to the instructions of the ELISA kits (catalog number: 47–37–01–15–01 Shanghai Jingmei Industrial, Shanghai, China).

### 2.6. Assay of Antioxidant Enzyme Activities in the Ileum

The activities of total superoxide dismutase (T-SOD), glutathione peroxidase (GSH-Px), glutathione S-transferase (GST) and malondialdehyde (MDA) in the ileum were all ascertained by using assay kits with absorbance measurement. The assay kits used in this study were ordered from Nanjing Jiancheng Bioengineering Institute (Nanjing, China).

### 2.7. Histopathological Analysis of Ileum

The ileum was accurately positioned relative to the crypt and the villus axis during the paraffin embedding process. The ileum sections (thickness, 5 μm) were carried out with standard pretreatment and stained with hematoxylin-eosin for review and analysis. Eight visual fields of each sample were randomly selected and observed. Histological morphometric variables were analyzed with panoramic MIDI (3D Histech, Budapest, Hungary), quantified and photographed. Eight villi per bird were measured with an analysis software (NIH Image J system, Bethesda, MD, USA).

### 2.8. Ultrastructural Morphology Analysis of Ileum

Ileal tissue ultrastructural morphology was observed and classified by transmission electron microscopy (TEM). After fixation, the tissues were dehydrated with different gradients of ethanol (50%, 70%, 90%, 100%) for 10 min. The tissues were embedded in a mixture of Araldite and Epon. Ultrathin sections (100 nm) were cut on an ultramicrotome (EM UC6, Leica, Munich, Germany). Finally, 15 visual fields of each sample were randomly selected and the representative images were taken on with a digital electron microscope (Hitachi S–4800, Shimadzu, Tokyo, Japan).

### 2.9. Quantitative Real-Time PCR (qRT–PCR)

RNA isolation was conducted as previously reported [22]. All PCR primers were ordered from Sangon Biotech (Shanghai, China) and the sequences are listed (Supplemental Table S2). RT–qPCR was performed in the same 96-well PCR plate and run at the same time with two repetitions per sample. The primers and cDNA were mixed and carried out by an amplification System (Monad q225, Monad, Suzhou, China). The levels of relative genes were detected using the $2-\Delta\Delta Ct$ method and qualitative comparative compared with β-actin expression.

### 2.10. Immunoblot Analysis of Ileum Protein

Immunoblot analysis was conducted as previously described in our laboratory [22]. The supernatant of ileum samples was obtained, and protein was extracted with the lysis. Then, the protein concentrate was mixed with an equal volume of loading buffer (catalog number: P0015, Beyotime, Shanghai, China) and separated under a definite voltage (120 V, 70 min) with sodium dodecyl sulfate-polyacrylamide gel electrophoresis (SDS–PAGE) (5% concentration gel and 12% of separation gel). Images of the blots were obtained and adjusted by the Essential V6 virtual 2D imaging software (UVITEC, Cambridge, UK). All the experiments were repeated 3 times.

### 2.11. Statistical Analysis

The experimental data of each sample were obtained from eight measurements. Results are expressed as mean ± standard deviation (mean ± SD) and analyzed using Statistical Product and Service Solutions (SPSS, Version 22.0, SPSS Inc., Chicago, IL, USA). Statistical significance of the date was evaluated using ANOVA followed by a least significant difference (LSD) test as the post hoc test with 5% probability of error and a value of $p < 0.05$ was considered statistically significant. All the graphs with standard deviation bar were made by GraphPad Prism in this study (version 8.3.0, GraphPad Software, San Diego, CA, USA).

## 3. Results

### 3.1. Analysis of AFB$_1$-DNA Adducts in Serum

AFB$_1$-DNA adducts in the serum of the AFB$_1$ group were significantly increased compared with those of the CON group and RES+ AFB$_1$ group, ($p < 0.05$). The level of serum AFB$_1$-DNA adducts in the RES+ AFB$_1$ group was significantly higher than that in the CON group ($p < 0.05$) but significantly lower than that in the AFB$_1$ group ($p < 0.05$). (Table 1).

**Table 1.** Effect of RES on the AFB$_1$-DNA adducts level and the antioxidant capacity in the plasma of AFB$_1$-exposed ducks.

| Items | Groups | | |
|---|---|---|---|
| | **CON** | **AFB1** | **RES + AFB1** |
| AFB1-DNA adducts (ng/mg protein) Biomarker of pro-oxidative stress and antioxidant | 0.158 ± 0.01 [c] | 0.448 ± 0.01 [a] | 0.273 ± 0.02 [b] |
| T-SOD (U/mg mL) | 173.36 ± 11.09 [a] | 151.07 ± 12.66 [b] | 158.39 ± 9.68 [b] |
| GSH-Px (U/mg mL) | 53.64 ± 1.47 [a] | 37.04 ± 1.39 [c] | 46.46 ± 1.39 [b] |
| GST (U/mg mL) | 19.79 ± 1.55 [a] | 16.67 ± 1.38 [b] | 18.48 ± 0.78 [a] |
| MDA (nmol/mg mL) | 0.143 ± 0.01 [b] | 0.212 ± 0.01 [a] | 0.174 ± 0.01 [b] |

The values are shown as mean ± SD of 8 individual male duck. CON, control group; AFB$_1$, AFB$_1$ group; RES + AFB$_1$, RES + AFB$_1$ group. Values are expressed as mean ± SD, $n = 8$. T-SOD: Total superoxide dismutase; GSH-Px: Glutathione peroxidase; GST: Glutathione s-transferase; MDA: Malondialdehyde; Labeled (a, b, c, a > b > c) means in a row without a common letter differ, $p < 0.05$.

### 3.2. Analysis of Ileal Antioxidant Capacity

Compared with the CON group, the concentration of MDA in the $AFB_1$ group was increased significantly ($p < 0.05$), the activities of GSH-Px and GST were decreased significantly ($p < 0.05$), and the level of T-SOD was decreased significantly ($p < 0.05$). Moreover, compared with the $AFB_1$ group, the RES treatment significantly increased ($p < 0.05$) the activity of GST, and it significantly increased ($p < 0.05$) the activities of GSH-Px. The MDA concentration in the RES+ $AFB_1$ group was decreased significantly ($p < 0.05$) compared with that in the $AFB_1$ group. However, there was no significant difference in the activity of T-SOD in the ileum tissues between the $AFB_1$ group and the RES+ $AFB_1$ group. Notably, the levels of MDA and GST did not differ between the CON group and the RES+ $AFB_1$ group and the levels of T-SOD and GSH-Px in the RES+ $AFB_1$ group were significantly lower than those in the CON group ($p < 0.05$). (Table 1).

### 3.3. mRNA Expression of Cytochrome P450 in Duck Ileum

The mRNA expression levels of the CYP enzymes CYP1A1, CYP1A2, CYP2A6, and CYP3A4 were increased in the $AFB_1$ group compared with those of the CON group ($p < 0.05$). However, in the group of RES+ $AFB_1$, the mRNA levels of the CYP1A1, CYP1A2 and CYP2A6 genes ($p < 0.05$) were inhibited compared with those in the $AFB_1$ group. Compared with the CON group, there was no significant change in the mRNA expression levels of CYP1A2 and CYP2A6. (Figure 1).

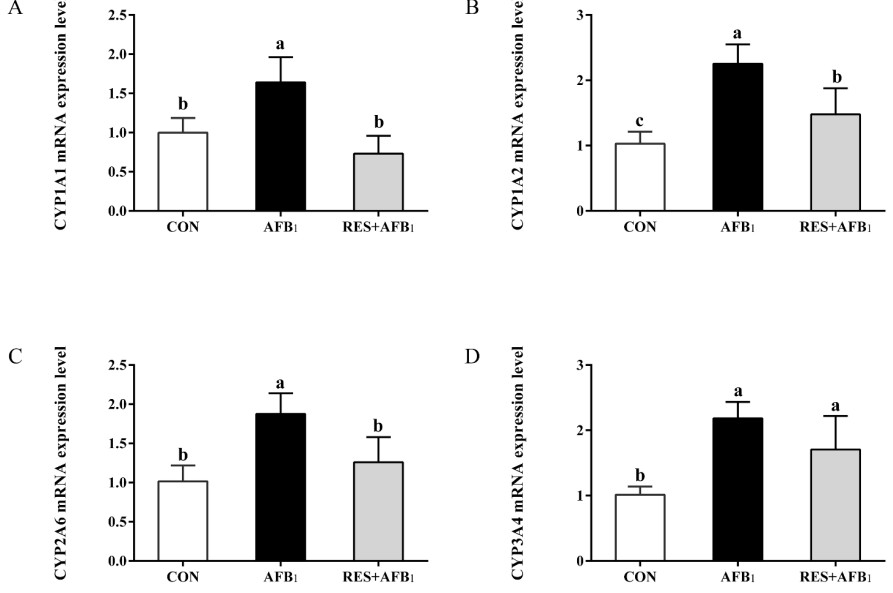

**Figure 1.** (**A–D**) Effect of RES on the mRNA expression levels of CYP enzymes in the ileum of $AFB_1$-exposed ducks. Values are expressed as mean ± SD, $n = 8$. Labeled (a, b, c, a > b > c) means in a row without a common letter differ, $p < 0.05$. CYP1A1: Cytochrome P4501A1; CYP1A2: Cytochrome P4501A2; CYP2A6: Cytochrome P4502A6; CYP3A4: Cytochrome P4503A4.

### 3.4. Histomorphological Changes in Ileum Tissues

Representative photomicrographs and ultrastructural pathological changes in ileum villi are shown in Figure 1. Compared with the CON group, histomorphological changes were observed in the ileum of the $AFB_1$-fed ducks, which included epithelial denudation (black arrow) and a corresponding decrease in villus height ($p < 0.05$). Microscopically, compared with the ileum of the $AFB_1$-fed ducks, 500 mg/kg RES supplementation in the diet increased villus height in the ileum, which was not different from that of the CON ducks unchallenged with $AFB_1$. As indicated by the white arrows, dietary RES alleviated epithelial denudation and repaired the structure of ileum villus compared with the $AFB_1$ group. However, there was no significant difference in crypt depth among

the CON group, the AFB$_1$ group and the RES + AFB$_1$ group. In comparison with the CON group, the AFB$_1$ group exhibited ultrastructural changes in intestinal microvilli and mitochondria, such as disappearance of mitochondrial cristae (black arrows pointed), and reduction in the microvillus height associated with larger gaps. In contrast, RES supplementation in diet restored fractured and swollen mitochondria with broken and cracked cristae induced by AFB$_1$. Meanwhile, RES supplementation in diet could restore the AFB$_1$-induced pathological changes in microvilli. (Figure 2).

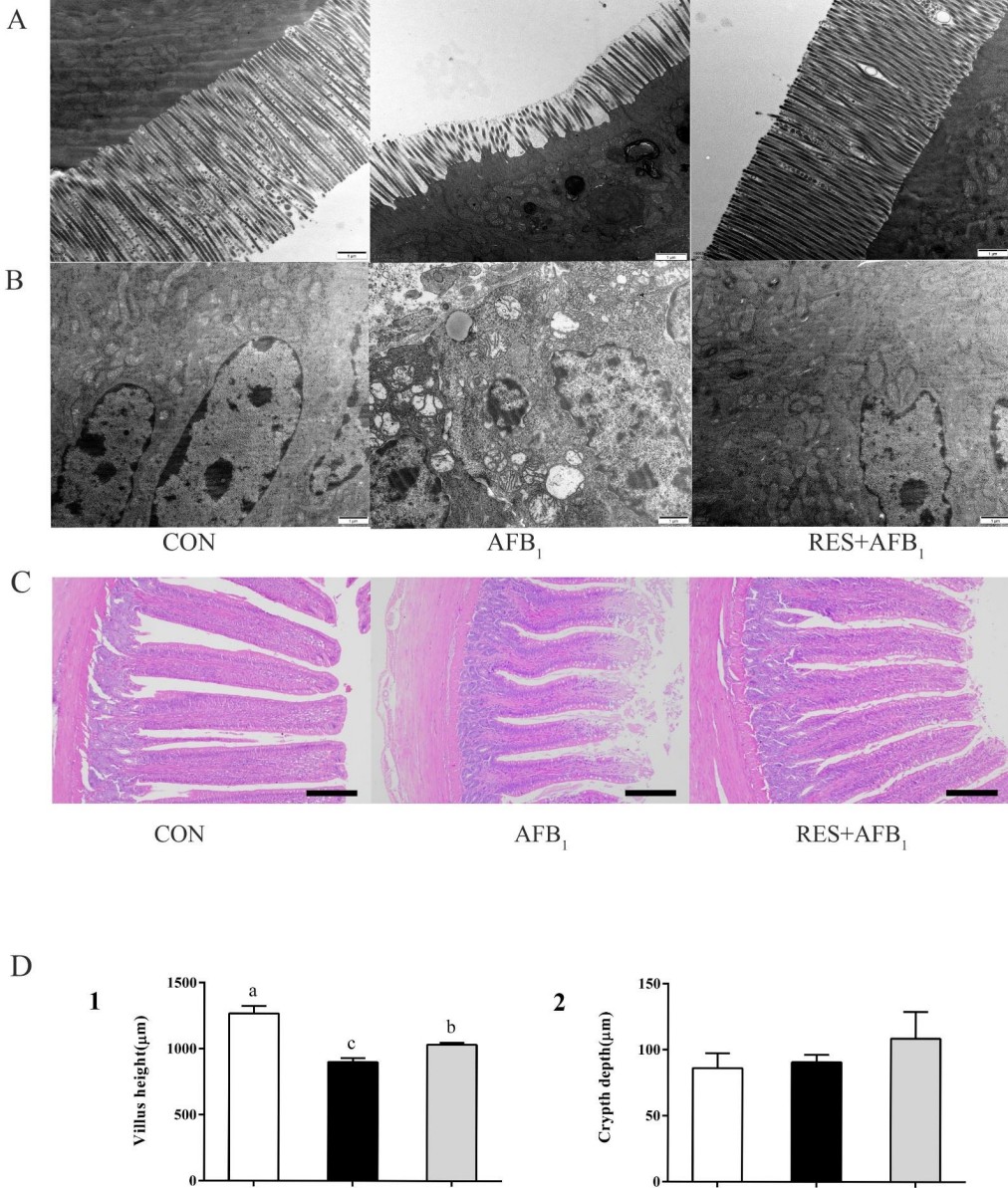

**Figure 2.** Effect of RES on histomorphological changes in ileum of AFB$_1$-exposed duck. (**A**): Ultrastructure of microvillus, (**B**): Cell and mitochondria in ileum mucosa. Original magnification 15,000×; scale bars, 1.0 μm). White arrowhead indicates the reduction in the micro villus height, and black arrowheads indicate vacuolization of mitochondria. (**C**): Representative photomicrographs of villi in ileum stained with hematoxylin-eosin (magnification, 200×; scale bars, 200 μm). Black arrowhead indicates epithelial denudation and the damaged villi structure. (**D**): (1) The villus height (2) The crypt elongation. Values are expressed as mean ± SD, *n* = 8. Labeled (a, b, c, a > b > c) means in a row without a common letter differ, *p* < 0.05.

### 3.5. mRNA and Protein Levels of Antioxidant-Related Genes

Compared with the CON group, the mRNA expression of Nrf2 was decreased significantly ($p < 0.05$) (Figure 3) and the protein expression of Nrf2 was decreased significantly in the AFB$_1$ group ($p < 0.05$) (Figure 4). Moreover, the mRNA and protein expression of Kelch-like ECH-associated protein 1 (Keap1) (Figures 3 and 4) was increased in the AFB$_1$ group. The mRNA levels of antioxidant genes (SOD1, CAT, GSH-Px, HO-1) and phase-II genes (GST, NQO-1, GCLM) (Figure 3) were significantly inhibited due to the toxicity of AFB$_1$. However, dietary RES increased the mRNA and protein expression of Nrf2 (Figures 3 and 4) caused by AFB$_1$ ($p < 0.05$). In addition, compared with the AFB$_1$ group, dietary RES significantly ($p < 0.05$) inhibited the gene expression of Keap1 (Figure 3). Compared with the AFB$_1$ group, the mRNA expression of protective genes which including GCLC, GCLM, HO-1, NQO1, SOD1, GSH-PX, GST and CAT (Figure 3) was increased in the RES+ AFB$_1$ group. Moreover, the mRNA levels of Nrf2, HO-1, SOD1 and NQO-1 did not differ between the CON group and the RES+ AFB$_1$ group (Figure 3).

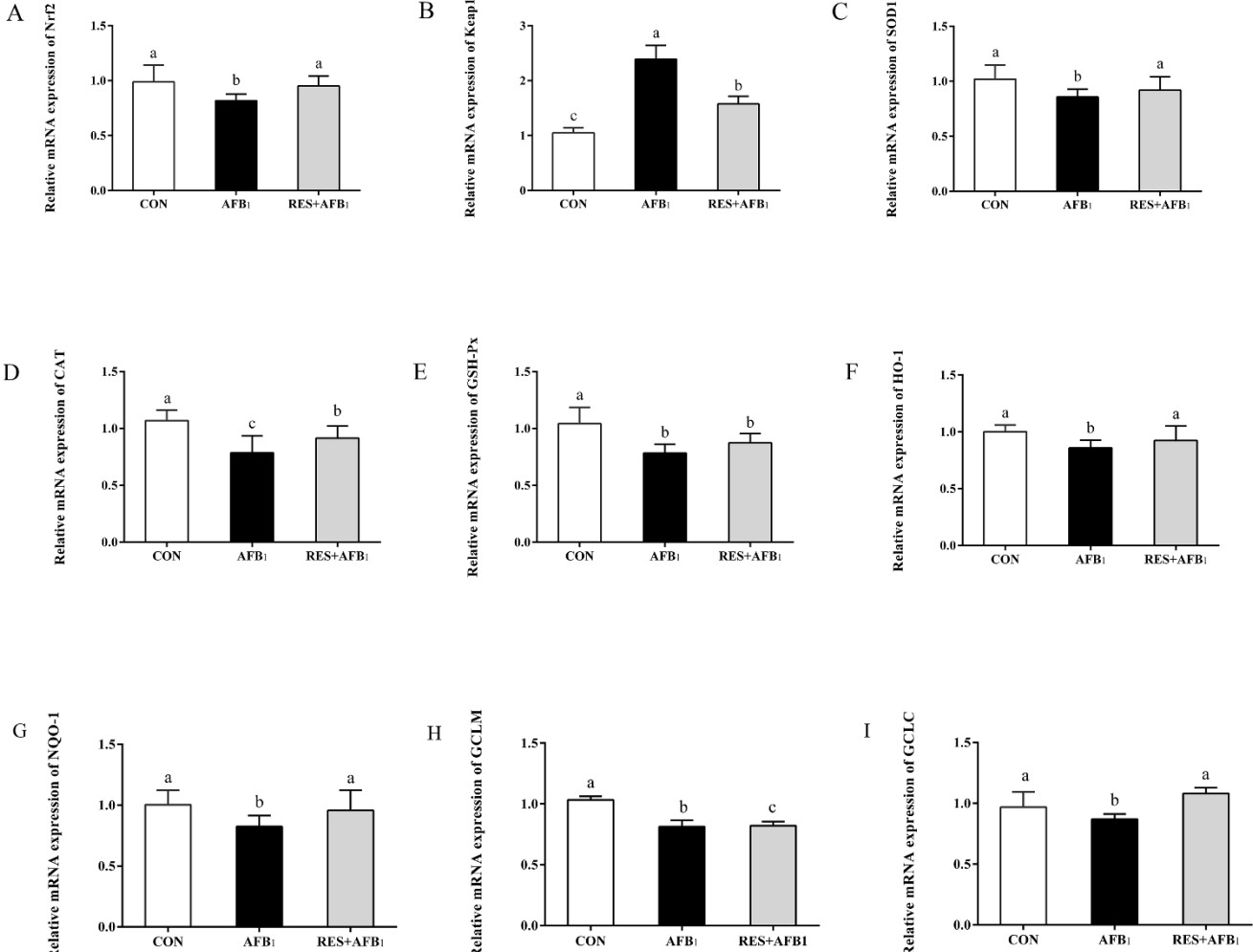

**Figure 3.** (**A–I**) Effect of RES on the mRNA expression levels of antioxidant genes in the ileum of AFB$_1$-exposed duck. Values are expressed as mean $\pm$ SD, $n = 8$. Labeled (a, b, c, a > b > c) means in a row without a common letter differ, $p < 0.05$. Nrf2: Nuclear Factor E2-related Factor 2; Keap1: Kelch-like ECH-associated protein 1; SOD1: Superoxide dismutase 1; HO-1: Heme oxygenase-1; NQO-1: NADPH quinineoxidoreductase-1; GCLM: Glutamate-cysteine ligase modifier subunit; GCLC: Glutamate-cysteine ligase modifier subunit.

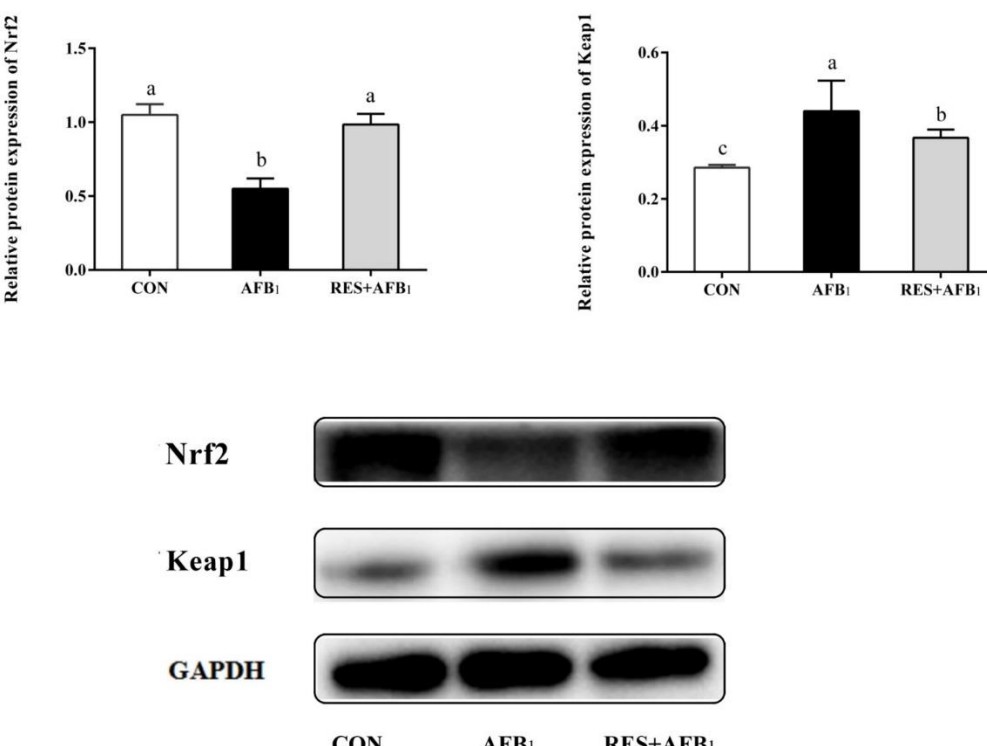

**Figure 4.** Effect of RES on protein expression of Nrf2 and Keap1 in the ileum of AFB$_1$-exposed duck. Values are expressed as mean $\pm$ SD, *n* = 8. Labeled (a, b, c, a > b > c) means in a row without a common letter differ, *p* < 0.05. Nrf2: Nuclear Factor E2-related Factor 2; Keap1: Kelch-like ECH-associated protein 1; GAPDH: Glyceraldehyde-3-phosphate dehydrogenase; CON: Control group; AFB$_1$: AFB$_1$ group; RES+ AFB$_1$: RES+ AFB$_1$ group.

*3.6. mRNA and Protein Levels of Inflammation-Related Genes*

The mRNA levels of inflammation-related genes, including NF-κB, NLRP3, TXNIP, IKK, P53, TNF-α, and IL-6 (Figure 5), in the AFB$_1$ group were significantly higher than those in the CON group. In addition, the protein levels of NF-κB, NLRP3 and TXNIP (Figure 6) in the ileum in the AFB$_1$ group were significantly increased compared with those in the CON group. However, RES significantly decreased (*p* < 0.05) the mRNA level changes in inflammation-related genes (Figure 5). The results were further verified by the downregulated protein expressions of NF-κB (*p* < 0.05), NLRP3 (*p* < 0.05) and TXNIP (*p* < 0.05) (Figure 6). Importantly, no changes were observed in the mRNA level of NF-κB or the protein levels of NF-κB and NLRP3 between the CON group and the RES+ AFB$_1$ group. Furthermore, the mRNA levels of NLRP3, TXNIP, IKK, P53, IL-6, IL-18 and TNF-α and the protein levels of TXNIP in the RES+ AFB$_1$ group were significantly lower than those in the CON group (*p* < 0.05).

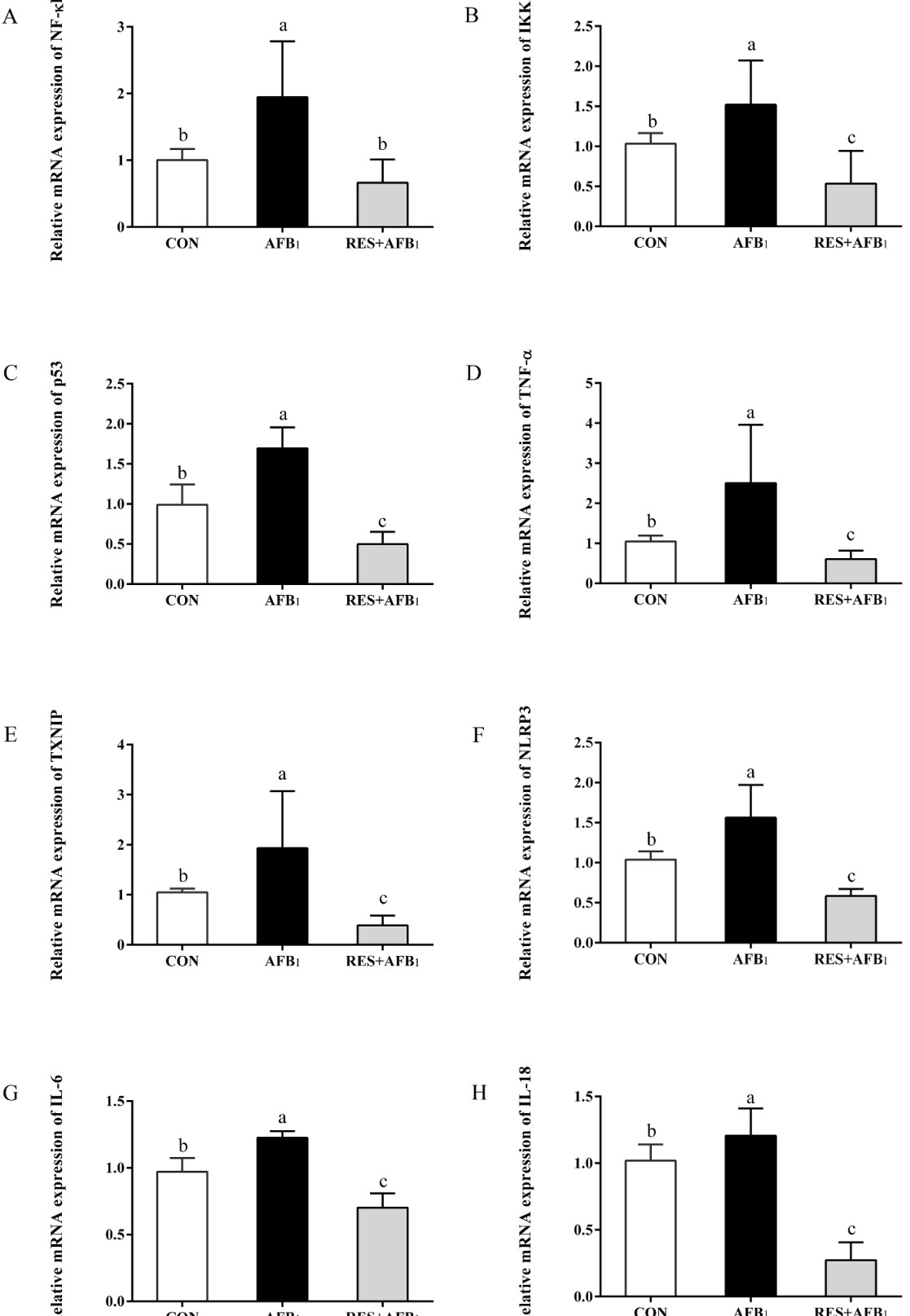

**Figure 5.** (**A**–**H**) Effect of RES on the mRNA expression levels of antioxidant genes in the ileum of AFB$_1$-exposed duck. Values are expressed as mean $\pm$ SD, *n* = 8. Labeled (a, b, c, a > b > c) means in a row without a common letter differ, *p* < 0.05. IKK: Inhibitor of nuclear factor kappa-B kinase; p53: p53 tumor suppressor protein; TNF-$\alpha$: Tumor necrosis factor-$\alpha$; IL-6: Interleukin-6; IL-18: Interleukin-18.

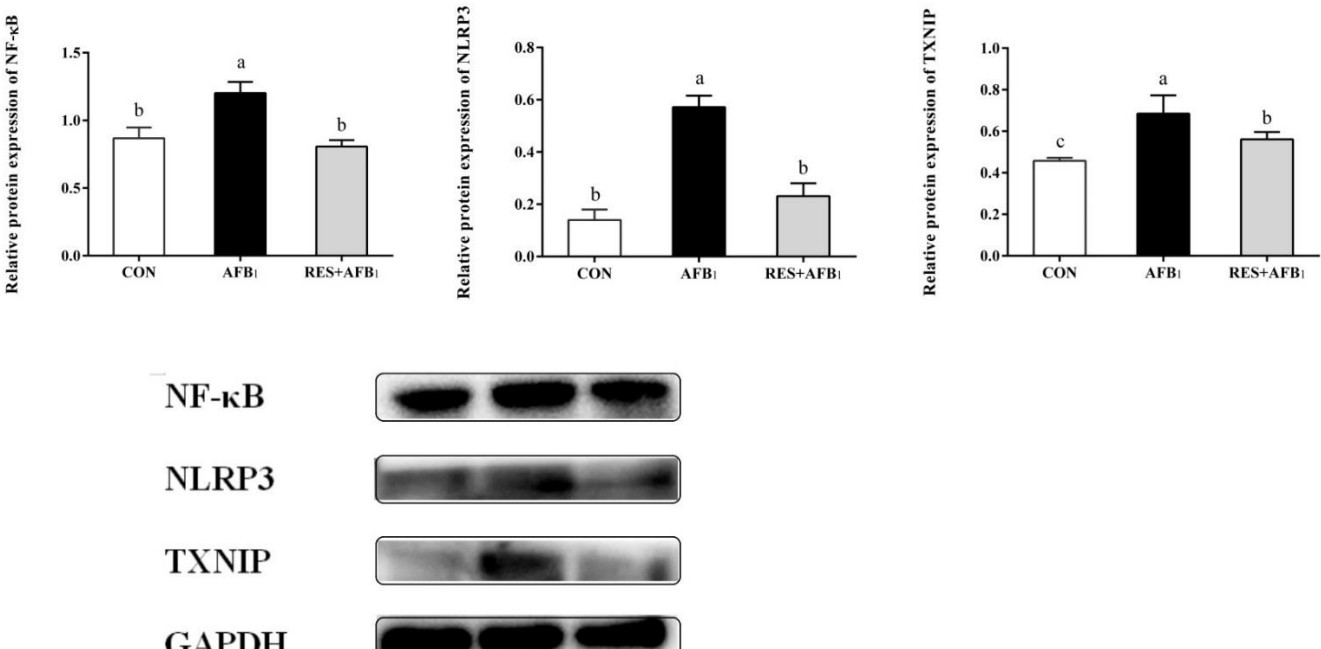

**Figure 6.** Effect of RES on protein expression of NF-κB, NLPR3 and TXNIP in the ileum of AFB$_1$-exposed duck. Values are expressed as mean $\pm$ SD, *n* = 8. Labeled (a, b, c, a > b > c) means in a row without a common letter differ, *p* < 0.05. NF-κB: Nuclear factor kappa B; NLRP3: Nod-like receptor families pyrin domain containing 3; TXNIP: Thioredoxin interacting protein.

## 4. Discussion

AFB$_1$, as one of the secondary metabolites produced mainly by molds, widely exists in animal feed and various kinds of agricultural products [23]. Intestinal injury induced by AFB$_1$ has been investigated in previous research, such as AFB$_1$-induced pathological changes and villus morphology changes in chicken ileum [24]. In fact, there are few reports about the toxicity of AFB$_1$ on the ileum of ducks. In this study, we established an in vivo AFB$_1$-exposure duck model to explore the enterotoxicity of AFB$_1$ on the ileum of ducks, and the protective effects of RES as an antioxidant in the diet on AFB$_1$-induced ileum injury in ducks.

Increasing evidence suggests that AFB$_1$ induces inflammation and oxidative damage in intestinal tissues, and leads to morphological lesions [25,26]. Meanwhile, RES has been reported to protect the intestinal morphology in weaning piglets, due to its antioxidant capacity [27]. The harmful effect of AFB$_1$ on the intestinal villus architecture was investigated, and villus blunting and epithelial denudation of the ileum in the AFB$_1$ group was observed. However, the changes were reduced in the RES+ AFB$_1$ group. These outcomes indicate that dietary RES alleviated the intestinal damage induced by AFB$_1$ and increased villus height. Mitochondrial dysfunction and cell apoptosis were associated with the oxidative stress caused by AFB$_1$. Mitochondria are dynamic organelles with important metabolic and regulatory functions, and the crista in mitochondria is regarded as an important structure to maintain the integrity and function of mitochondria. Cao et al., 2009 demonstrated that supplementation in a diet with RES restored the diquat-induced breaking of cristae, vacuolization and turgidity of the mitochondrion by decreasing ROS level in vivo [28]. In addition, the ultrastructure of mitochondria in the ileum was impaired, as noted by the presence of vacuolization and turgidity, which was greatly restored by RES with the majority of cristae having regular shapes. These qualitative results revealed that RES could alleviate AFB$_1$-induced mitochondrial damage.

The CYP450 enzyme system plays a fundamental role in the activation of AFB$_1$. The activation of CYP450 enzymes can induce and enhance the genotoxicity of AFB$_1$ [17]. Diaz et al., 2010 found that the high sensitivity of ducks to AFB$_1$ toxicity is due to the

involvement of four enzymes, CYP1A1/2, CYP2A6 and CYP3A4, in the conversion of AFB$_1$ to AFBO [29]. Subsequently, AFBO binds to DNA at guanine residues and forms a variety of AFB$_1$-DNA adducts [4]. Shao et al., 2019 reported that AFB$_1$-DNA adducts can bind to nucleoproteins and nucleic acids and that decreased levels of antioxidant enzymes further induce DNA damage, cell injury and disturb protein synthesis [30]. Natural plant polyphenols have been reported to inhibit CYP activity and reduce the production of AFB$_1$-DNA adducts by increasing the levels of antioxidant enzymes (GSH-Px, SOD, CAT and GST) [31]. This result is similar to previous studies, which found that AFB$_1$ toxicity promoted the activation of CYP1A1, CYP1A2, CYP2A6 and CYP3A4 and significantly increased the production of AFB$_1$-DNA adducts. However, the mRNA expression of CYP1A1 and CYP1A2 genes was reduced significantly by RES due to the antioxidative ability of RES. As a potential inhibitor, dietary RES involved in alleviating DNA damage and reducing the production of AFB$_1$-DNA adducts in the ileum of ducks via the expression levels of CYP1A1 and CYP1A2.

　　As the principal inhibitor of the Nrf2, Kelch-like ECH-associated protein 1 (Keap1) plays a pivotal role in the balance of intracellular redox homeostasis [32]. Under normal conditions, Keap1 and Nrf2 form a complex in the cytoplasm and induce the inhibition of the Nrf2 signaling pathway. A previous study suggested that oxidative stress is an inevitable consequence of AFB$_1$-induced toxicity [33]. This oxidative stress induced the dissociation of Keap1 from Nrf2/Keap1 [34]. When the dissociation of Keap1-Nrf2 complexes occurred, Nrf2 was recognized and transported to the nucleus, then efficiency bound to the regulatory sites of DNA and induced the activation of protective enzyme system. Nrf2 was suggested, which could regulate antioxidant response elements (AREs) and upregulate the downstream antioxidation genes, including SOD, catalase (CAT), GSH-Px, heme oxygenase-1 (HO-1), GST, NQO1, glutamate-cysteine ligase catalytic (GCLC) and GCLM [35]. A previous study noted that GSH-Px and T-SOD counterbalanced the homeostasis of oxidation and antioxidation. Two classes of detoxification enzymes involved in the detoxification, which included phase-I and phase-II enzymes. Phase I enzymes modify the structures of foreign compounds via oxidation, reduction, or hydrolyzation reactions. Phase II enzymes are involved in the metabolism and detoxification of environmental carcinogens [36]. As a class of phase-II detoxifying enzymes, GST, NQO1 and GCLM are involved in various detoxification processes in vivo and plays a vital role in cell protection and relieving oxidative stress [37–39]. The results of this study indicated that dietary RES significantly increased antioxidant enzyme (GSH-PX, GST and T-SOD) activity, and reduced the MDA content in ileum of ducks. Consistently, Zhang et al., 2003 suggested that RES alleviated cadmium-induced reduction in antioxidant capacity by reducing the level of MDA and activating the activities of antioxidant enzymes in a chicken [40]. Oxidative stress induced by the generation of ROS and lipid peroxidation is the main cause of AFB$_1$-induced toxicity [41]. In this study, AFB$_1$ strongly suppressed the activation of Nrf2, then subsequently induced the inhibition of antioxidative proteins by reinforcing the sequestration of Keap1 and Nrf2. Importantly, the mRNA and protein level of Keap1 was increased significantly in the AFB$_1$ group, which showed that AFB$_1$ negatively regulated the Nrf2 signaling pathway. Moreover, AFB$_1$ significantly inhibited the mRNA expression of these protective genes (GCLC, GCLM, HO-1, NQO1, SOD, GSH-Px, GST and CAT). This study found that AFB$_1$-induced enterotoxicity caused oxidative stress and induced the inhibition of the Nrf2 signaling pathway by increasing the mRNA and protein expression levels of Keap1. However, the intestinal oxidative stress caused by AFB$_1$ can be relieved by RES by activating the Nrf2/Keap1 signaling pathway.

　　Recent research has shown that oxidative stress and inflammation are highly related [42]. A previous study noted that AFB$_1$ upregulated the level of NF-κB and upregulated the content of many inflammatory cytokines [43]. NLRP3 was considered to be a key target in many systems and organs of animals and it can upregulate Caspase-1 mRNA levels and induce the synthesis and metabolism of IL-1β and IL-18, thus leading to inflammation and damage [44]. NLRP3 is an important inflammasome in both animals

and humans that can be regulated by activating NF-κB signals and binding with TXNIP to identify many pathogens and trigger the activation of immune system [45]. In this study, the mRNA levels of IL-1β, IL-6, IL-18 and TNF-α were increased in the AFB$_1$ group, and returned to normal levels in the RES+ AFB$_1$ group. The results in present study indicate that NF-κB mRNA and protein were significantly increased in the AFB$_1$ group. Dietary RES decreased both the mRNA and protein levels of NF-κB in the ileum of AFB$_1$-damaged ducks. This result suggested that NF-κB was a molecular target of AFB$_1$-induced inflammation, which was in agreement with a previous study [46]. Meanwhile, Szkudelska et al., 2020 reported that RES could prevent the increase in IL-6, IL-1β, TNF-α by inhibiting the expression of NF-κB in the skeletal muscle of diabetic rats [47]. In order to gain insight into the mechanism by which dietary RES reduces the AFB$_1$-induced inflammation, the expression levels of TXNIP, NLRP3, and Caspase-1 were also detected to evaluate the status of the NLRP3 mediated inflammatory pathway. The mRNA and protein expressions of TXNIP and NLRP3 were increased in ducks fed AFB$_1$ and could be reversed by dietary RES. Furthermore, the protein level of Caspase-1 was increased by AFB$_1$ induction, and the mRNA levels of downstream inflammatory genes including IL-6 and IL-18 and the content of inflammatory cytokines in the ileum were increased. These results demonstrated that RES alleviated the AFB$_1$-induced duck ileitis by reducing AFB$_1$-induced oxidative stress, therefore inhibiting the activation of the NF-κB signaling pathway and TXNIP/NLRP3 signaling pathway.

## 5. Conclusions

AFB$_1$ induced apoptosis, mitochondrial dysfunction and ileal injury in ducks mediated by oxidative stress, DNA damage, and inflammation. RES supplementation in the diet effectively reduced the enterotoxicity caused by AFB$_1$, such as reducing lipid peroxidation and improving intestinal barrier and mitochondrial function. In addition, the expression of CYP450 enzymes induced oxidative stress via the Nrf2/Keap1 signaling pathway. RES decreased the production of AFB$_1$-DNA adducts by downregulating the expression of CYP1A1 and CYP1A2 and reduced DNA damage and oxidative stress by regulating the Nrf2/ Keap1 and NF-κB/NLRP3 signaling pathways. The results of this study will help to provide greater insight into the molecular mechanism of dietary RES involved in alleviating intestinal injury induced by mycotoxins.

**Supplementary Materials:** The following are available online at https://www.mdpi.com/article/10.3390/agriculture12010054/s1, Table S1: Ingredients and nutrient composition of basal diet (on an air-dried basis), Table S2: Primer sequences and productlengths of target genefragments.

**Author Contributions:** The experimental work was performed under first author and corresponding author. H.Y. as first author of this manuscript prepared the first draft. In the process of feeding experiment, S.J. made an important contribution. Most of the experimental analysis was completed by Y.J. and C.Y. During the data processing stage, H.Y. and Y.W. presided over this part of the work. R.Z. participate in the collation of references and reviewed the first draft. X.F. revised for the final version. All authors have read and agreed to the published version of the manuscript.

**Funding:** The whole study took 2 years and funded by National Natural Science Foundation of China (32072768, 31772638).

**Institutional Review Board Statement:** All experimental procedures in our study followed the guidelines of the Declaration of Helsinki, and the Northeast Agricultural University Institutional Animal Care and Use Committee (Protocol number: NEAU (2011)-9) approved and fully agreed with the study.

**Informed Consent Statement:** Not applicable.

**Data Availability Statement:** Not applicable.

**Acknowledgments:** We thank Xingjun Feng for comments, support and direction and thank Anshan Shan and Xiao Liu for their advice and support.

**Conflicts of Interest:** There is no potential interest relationship in the process of publishing the manuscript.

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
