# Peer review of "Dietary Resveratrol Alleviates AFB1-Induced Ileum Damage in Ducks via the Nrf2 and NF-κB/NLRP3 Signaling Pathways and CYP1A1/2 Expressions"

_agriculture, doi:10.3390/agriculture12010054_

Round 1

Reviewer 1 Report

The authors evaluated the Dietary resveratrol alleviates AFB1-induced ileum damage in ducks by regulating the Nrf2 and NF-κB/NLRP3 signaling pathways and CYP1A1/2 expressions.

Some minor remarks are follow.

Figure 2, 3. Please write the captions on the axes correctly. 

Line 429. Supplementary Materials. Please put this materials in a separate field, according the journals details.

Reference 3 is incorrect. Please write it again.

Reviewer 2 Report

The paper is written well with minimal grammar or language issues and has interesting scientific findings. However, major issues with the paper are listed below:

  • The current design lacked a control group that received RES without exposure to AFB1 to determine the efficacy of RES in ducks in such dose.

  • Why did the authors choose this dose of 500 mg/kg of RES against AFB1, please add references for the used levels of RES and AFB1.

  • Line 118: RES-containing and AFB1-containing feed was made by mixing in powder form: please rewrite this sentence.

  • Line 121: eight ducks in three groups were randomly selected for samples collection; please determine no of samples per group (how eight ducks were selected, and each group contained 5 replicates)

  • Line 121: The samples were collected following feed withdrawal a 12 hour, how this fasting period did not alter the histomorphological changes in ileum tissues and consequently the obtained results.

  • Please correct the position of fig 2 to be vertical.

  • Please adjust all decimal points to be the same in all tables (two decimal points).
  • Authors should define all abbreviations in all tables and figures footers and captions ( AFB1, RES+AFB1, SOD1, NQO-1 ……..…).

Reviewer 3 Report

This manuscript by Wang et al illustrates how dietary resveratrol (RES) can reduce aflatoxin B1 (AFB1) induced ileal damage in ducks.

The manuscript idea is a continuation of 3 recently published articles by the same research group published in 2021,

Yang, Hao, et al. "Dietary resveratrol alleviated lipopolysaccharide‐induced ileitis through Nrf2 and NF‐κB signalling pathways in ducks (Anas platyrhynchos)." Journal of Animal Physiology and Animal Nutrition (2021).

Jin, Sanjun, et al. "Effects of dietary resveratrol supplementation on the chemical composition, oxidative stability and meat quality of ducks (Anas platyrhynchos)." Food Chemistry (2021): 130263.

Jin, Sanjun, et al. "Dietary supplementation of resveratrol improved the oxidative stability and spatial conformation of myofibrillar protein in frozen-thawed duck breast meat." Food Bioscience 43 (2021): 101261.

Major issues:

The scanning electron microscopy step is totally wrong, the authors are showing transmission electron microscopic images and the explanation of this step in material and methods is a mixture of both techniques which is not correct and indicates that the authors have no experience in SEM or TEM. And the explanation of the TEM findings in the results section is not correct, the authors should seek the help of an expert in interpreting TEM images.

Gene expression results are shown as numerical values in tables (Table 3 and Table 4) which makes it difficult to understand, it must be presented as bar graphs.

The figures are of low-quality, high-quality images are necessary especially the histopathological images and TEM images

Comments:

Abstract:

Line 12: “in Aflatoxin B1-induced ducks” … the sentence is not clear, are the ducks are not induced by AFB the ileum injury is induced, please revise.

Introduction:

Line 33: “energy feed for poultry feed” … please revise, the sentence is not clear.

Line 49: “including immune adjunction” … please revise, this expression is not correct.

Material and methods:

Lines 106-110: are repeated from the previous point 2.2., please remove.

Line 125: “Ducks were anesthetized” … please explain the type and method of anesthesia and the agent used.

Lines 127-129: “The ileum sample for scanning electron microscopy (SEM) was stored in the electron microscope immediately frozen in liquid nitrogen and then stored at -80 °C” … these lines are not clear. What do you mean by “was stored in the electron microscope immediately” ??

Line 138: “SPSS” … please define.

Line 139: “T-SOD, GSH-Px, MDA” … first mention, please define.

Results:

Lines 215-216: please revise the sentence, the reported results in it are not correct.

Line 237 “villus blunting” and line 238 “decrease villus height” … please explain the difference.

Lines 249-250: please revise the sentence is not clear.

Line 255: the scale bar on the images is 1 µm,

Line 267: phase II genes … first mention of this statement .. please explain in material and methods or discussion.

Discussion:

Lines 360-362: there is no evidence that AFB1 can result in reactive electrophiles .. this point of the discussion needs revision.

Round 2

Reviewer 2 Report

All required issues were resolved

Reviewer 3 Report

All required revisions were done, no further comments are required.